# Angle-Controlled Nanospectrum Switching from Lorentzian to Fano Lineshapes

**DOI:** 10.3390/nano14231932

**Published:** 2024-11-30

**Authors:** Fu Tang, Qinyang Zhong, Xiaoqiuyan Zhang, Yuxuan Zhuang, Tianyu Zhang, Xingxing Xu, Min Hu

**Affiliations:** 1Terahertz Research Center, School of Electronic Science and Engineering, University of Electronic Science and Technology of China, Chengdu 610054, China; 202112022427@std.uestc.edu.cn (F.T.); qinyangzhong1225@163.com (Q.Z.); 2022020912026@std.uestc.edu.cn (Y.Z.); tyzhang@uestc.edu.cn (T.Z.); 202011022904@std.uestc.edu.cn (X.X.); hu_m@uestc.edu.cn (M.H.); 2Key Laboratory of Terahertz Technology, Ministry of Education, Chengdu 610054, China

**Keywords:** spectral lineshapes, fano profiles, near-field optical microscopy (s-SNOM), nanoscale phase sensors

## Abstract

The tunability of spectral lineshapes, ranging from Lorentzian to Fano profiles, is essential for advancing nanoscale photonic technologies. Conventional far-field techniques are insufficient for studying nanoscale phenomena, particularly within the terahertz (THz) range. In this work, we use a U-shaped resonant ring on a waveguide substrate to achieve precise modulation of Lorentzian, Fano, and antiresonance profiles. THz scattering scanning near-field optical microscopy (s-SNOM) reveals the underlying physical mechanism of these transitions, driven by time-domain phase shifts between the background excitation from the waveguide and the resonance of the U-shaped ring. Our approach reveals a pronounced asymmetry in the near-field response, which remains undetectable in far-field systems. The ability to control spectral lineshapes at the nanoscale presents promising applications in characterizing composite nanoresonators and developing nanoscale phase sensors.

## 1. Introduction

The Lorentzian resonance arises from the exponential decay of a discrete excited state with a finite lifetime, resulting in symmetric lineshapes. The Fano lineshape arises from the interference between a discrete resonant state and a continuum, resulting in an asymmetric profile [1]. Manipulating the spectral lineshapes between Lorentzian and Fano profiles not only enables the design of multifunctional photonic devices for applications such as optical sensing [2], switching [3], filtering, and modulation [4,5,6], but also reveals intriguing quantum effects [7], including exceptional point phenomena [8] and complex resonant state interactions [9], which have been widely studied in nuclear and solid-state physics. These effects are crucial for practical applications in photonic devices and for advancing fundamental understanding of quantum interactions. In the THz range, these phenomena have primarily been explored through far-field systems, which lack nanoscale resolution. By tuning Lorentzian and Fano lineshapes at the nanoscale, we can uncover fundamental physical processes with unmatched precision, providing valuable insights for characterizing nanoscale materials and devices, thereby deepening our understanding of nanoscale interactions [10,11,12]. Consequently, there is an urgent need for new methods to achieve such spectral control.

THz scattering scanning near-field optical microscopy (s-SNOM) uses a cantilevered metallic tip to focus incident radiation into a nanoscale volume, enabling imaging and spectroscopy beyond the diffraction limit. This exceptional spatial resolution and low photon energy make THz s-SNOM ideal for studying phase transitions [13], carrier dynamics [14], high-sensitivity sensing [15], and subwavelength-confined modes like plasmons and phonons [16], providing insights unattainable with conventional far-field techniques [17]. Furthermore, the highly localized field excited by the tip facilitates precise control of nanoscale optical properties, enabling targeted manipulation of resonant behaviors [18].

Here, we demonstrate tunable transitions between Lorentzian and Fano lineshapes at the nanoscale using a single U-shaped resonant ring on a waveguide substrate, achieved by rotating the structure under THz s-SNOM. This transition is driven by the relative phase shift in the time domain between the background excited by the waveguide substrate and the ring resonance. The tip-excited localized near-field further enables control over nanoscale spectral lineshapes, revealing an asymmetry in the lineshapes transition induced by the tip, which is not observable in far-field measurements. Such unique asymmetry is crucial for understanding near-field behaviors and tuning the optical response of nanoscale resonant structures, providing insights into underlying physical mechanisms and enabling advanced control over light-matter interactions in nanophotonic applications.

## 2. Materials and Methods

### 2.1. Experimental Setup

In our experiment, we employed a commercial scattering-type terahertz time domain spectroscopy near-field scanning system (THz-TDS s-SNOM) developed by Nearspec, Haar, Germany. This system integrates Menlosystem’s terahertz time-domain spectrometer with Nearspec’s near-field AFM mainframe, forming a high-precision terahertz near-field scanning imaging platform that provides optimal experimental conditions for high-resolution terahertz spectroscopy and imaging studies.

The terahertz time-domain spectrometer system utilizes a high-performance 1560 nm erbium-doped fiber femtosecond laser to excite an InGaAs/InAlAs superlattice heterojunction antenna, generating terahertz radiation. Figure 1 illustrates the overall system layout. The key parameters of the femtosecond laser include a repetition rate of 100 MHz, an average output power of 100 mW, a pulse width of 45 fs, and a 6 m dispersion-compensated fiber. These parameters are critical for generating high-quality terahertz pulses and ensuring that the pulses retain their temporal characteristics during transmission. The laser output is split into two beams, each with an average power of 50 mW. One beam, after power attenuation, reaches the terahertz emitting antenna with an average power of less than 30 mW, while the other beam passes through a delay line and reaches the terahertz receiving antenna. The laser pulses arriving at the transmitting and receiving antennas are broadened to a pulse width of 100 fs after passing through the 6 m dispersion-compensated fiber, enabling effective excitation and detection of terahertz pulses.

In the experiment, the pulses generated by the terahertz antenna were not initially collimated. Therefore, an integrated hyper-hemispherical silicon lens was used at the antenna end to collimate the divergent terahertz pulses. The use of the lens significantly improved the spatial characteristics of the terahertz radiation, facilitating better transmission and focusing onto the target region. The P-polarized terahertz pulses generated by the system are focused onto the interaction region between the atomic force microscope (AFM) tip and the sample through a custom off-axis parabolic mirror after being reflected by a prism. The off-axis parabolic mirror has a focal length of 16 mm, and the incidence angle of the terahertz pulses is 52°. This design ensures high precision at the focal point and effectively reduces aberrations.

During the near-field scanning process, the AFM mainframe operates in tapping mode, which minimizes damage to the sample surface while maintaining high-resolution imaging. In tapping mode, the near-field signal is confined to the local region between the AFM tip and the sample surface, modulated by the resonant frequency of the tip, and subsequently scattered into the far field, where it is collected and collimated by the off-axis parabolic mirror before reaching the receiving antenna. This approach allows the system to effectively capture electromagnetic information near the sample surface, enabling imaging of the sample’s nanostructure.

During the transmission of the near-field signal from the surface to free space, significant background signal interference occurs, which adversely affects the signal’s purity. To obtain a cleaner, higher-order near-field signal, point-by-point scanning and integration are required to improve the signal-to-noise ratio. In traditional far-field detection, the detected signal at each scanning point remains relatively stable; however, in the near-field regime, each point’s scanning signal exhibits inherent instability due to background signal interference. Consequently, lock-in amplification is employed to modulate and demodulate the probe’s resonant frequency, with modulation frequencies being integer multiples of the probe’s resonance frequency. This modulation-demodulation method effectively extracts weak near-field signals, which are then amplified by the receiving antenna. By adjusting the scanning length of the delay line, near-field time-domain spectral reconstruction can be achieved, providing comprehensive information about both the surface and internal structures of the sample.

The probe used in this experiment is specifically designed for terahertz near-field optics, with model number 25PtIr500B-H50 from RMN. The cantilever length of the probe is 500 µm, and the tip length is 80 µm, with a resonance frequency of approximately 16 kHz, which corresponds to the operating frequency of the tapping mode. Due to variations in manufacturing quality, the resonance frequencies of different probes may exhibit subtle differences. The probe amplitude is 100 nm, optimized to balance sensitivity and stability. The design and operational parameters of the probe are crucial for ensuring effective modulation of near-field signals and stable interactions with the sample.

To reduce the influence of water vapor absorption on the terahertz spectrum during the experiment, the entire system was placed in a nitrogen environment with a humidity level below 5%. Water vapor exhibits strong absorption characteristics in the terahertz frequency range, which can significantly affect the quality of spectral data. By controlling the environmental humidity, we effectively minimized the interference from water vapor absorption, thereby enhancing the accuracy and reliability of terahertz measurements. Additionally, the use of a nitrogen environment helps stabilize the overall operating conditions of the experimental system, reducing the impact of external environmental fluctuations on the experimental results.

In summary, this experiment successfully constructed a high-resolution imaging system capable of finely probing nanoscale structures by integrating a high-performance terahertz time-domain spectrometer with a near-field AFM mainframe. The key optical components in the system, including the hyper-hemispherical silicon lens and off-axis parabolic mirror, ensured effective collimation and focusing of terahertz pulses, maximizing the interaction between the sample and terahertz radiation. The tapping mode employed by the near-field AFM, combined with lock-in amplification, enabled high-fidelity extraction of weak near-field signals. By precisely controlling experimental parameters, such as humidity and probe modulation, we achieved reliable measurements of complex material responses in the terahertz frequency range, providing strong experimental support for the study of optical and electrical properties of materials. This system demonstrates significant potential in terahertz near-field optical research, capable of revealing the microscopic structure and dynamic behavior of various nanomaterials and biological samples, which holds considerable importance for future materials science and biomedical research.

### 2.2. Material Preparation

Figure 1c illustrates a schematic of the sample as observed through an optical microscope. The precise dimensions of the U-shaped resonant ring structure in the inset are as follows: P = 22 µm, L = 30 µm, and W = 8 µm. In this experiment, the U-shaped resonant ring was fabricated via a photolithography process, followed by gold deposition using magnetron sputtering, resulting in a thickness of 100 nm. The substrate utilized was an SOI (Silicon-On-Insulator) silicon wafer, comprising three distinct layers: the top layer is high-resistivity silicon with a thickness of 20 ± 1 µm and a resistivity greater than 1000 Ω·cm; the intermediate layer is silicon dioxide with a thickness of 2 µm ± 5%; and the bottom layer is low-resistivity silicon with a thickness of 675 ± 10 µm and a resistivity in the range of 0.005 to 0.020 Ω·cm.

### 2.3. Data Processing

The normalization of terahertz near-field signals is highly sensitive to variations in sample surface roughness and temporal delay. Therefore, during the extraction of scattered near-field signals, it is standard practice to limit the scanned region to several tens of micrometers around the sample structure. This approach minimizes the time delay difference between the sample signal and the reference signal, as significant time delay differences can lead to a pronounced reduction in near-field signal strength, thereby introducing substantial errors in the normalization process.

To acquire near-field time-domain signals, an integration time of 200 ms per point is needed to ensure that the second-order near-field signal achieves a sufficient signal-to-noise ratio (SNR) for subsequent analysis. Under these conditions, third-order near-field signals are also detectable; however, their SNR ratio remains insufficient for meaningful data interpretation. The normalization process is described as follows [19,20]:(1)$${\mathrm{Norm}\;S}_{2}\left(\omega \right)=\left|\frac{S_{2,\mathrm{sample}}\left(\omega \right)}{S_{2,\mathrm{ref}}\left(\omega \right)}\right|$$
(2)$${\mathrm{Norm}\;P}_{2}\left(\omega \right)=P_{2,\mathrm{sample}}\left(\omega \right)-P_{2,\mathrm{ref}}\left(\omega \right)$$

Here, $S_{2,\mathrm{sample}}\left(\omega \right)$ and $P_{2,\mathrm{sample}}\left(\omega \right)$ are the amplitude and phase of the second-order near-field signal, respectively. Measurements are performed with the AFM tip scanning the surface of the sample in tapping mode. Similarly, $S_{2,\mathrm{ref}}\left(\omega \right)$ and $P_{2,\mathrm{ref}}\left(\omega \right)$ represent the amplitude and phase of the signal from the substrate surface, where the substrate material is high-resistivity silicon (HR-Si). During phase processing, baseline subtraction may be necessary to enhance data accuracy.

### 2.4. Method: Calculation of Fano Resonance Line Shape

To model the Fano resonance line shape and compute its corresponding absorption cross section in terms of frequency, we start with the standard Fano formula. The Fano resonance is characterized by the interaction between a discrete state and a continuum of states, leading to an asymmetric line shape in the absorption spectrum. The absorption cross section σ_Fano_(E) is given by the following expression:(3)$$\sigma_{\mathrm{Fano}}\left(E\right)=\sigma_{0}\frac{{\left(q+e\right)}^{2}}{1+e^{2}}$$
where: *E* is the photon energy, *q* is the Fano parameter that quantifies the asymmetry of the resonance, *e* = (*E−E*_0_)/(ℏΓ/2) is the normalized energy, with *E*_0_ being the resonance energy and Γ the width of the resonance, and *σ*_0_ is the cross section far from the resonance. 

## 3. Results

Adjacent U-shaped rings were spaced 500 μm apart to effectively prevent periodic interference between neighboring resonators, thereby ensuring independent analysis of each individual resonator. This strategic spacing eliminates potential cross-talk between the resonators, allowing for precise characterization of the resonant behaviors of a single structure without coupling effects. Optical and near-field time-domain white-light imaging results, presented in Figure 1b,c, demonstrate not only high spatial resolution but also a detailed macroscopic spectral response that can be observed across the sample surface. The ability to capture these high-resolution images is critical for understanding the resonant behavior at various spatial locations within the U-shaped ring.

The near-field white-light imaging was conducted using peak detection imaging mode, which focuses on maximizing the signal-to-noise ratio by capturing the strongest response at each frequency. While this imaging method provides excellent spatial resolution, it has inherent limitations in resolving resonance at specific frequencies, particularly those that may not be directly within the range of the detected signal or those that exhibit subtle frequency shifts. In Figure 2, we performed a near-field time-domain signal scan at a fixed point in the left arm of the U-shaped ring resonator, enabling precise tracking of temporal changes at a single location. The resonator was subsequently rotated incrementally from 0° to 90°, allowing us to systematically capture the variations in time-domain signals at different orientations. This careful rotation, coupled with time-domain analysis, allowed for an in-depth exploration of the resonator’s behavior as it responded to varying incident angles.

The resulting data from these scans were then subjected to Fourier transform analysis, which was used to derive the spectral information at each rotation angle of the U-shaped ring. Specifically, we extracted the real and imaginary components of the spectra at each orientation (Figure 2a,b). Importantly, throughout the entire scanning procedure, the measurements were consistently centered on the same point of the U-shaped resonator, ensuring that any variations observed were directly related to changes in the resonator’s orientation rather than shifts in the measurement position. Simulations confirmed that as the U-shaped ring rotated from 0° to 90° (shown in Figure 2c), the imaginary part of the spectral response transitioned from a Lorentzian-like lineshape (with the Fano parameter q→−∞) to a Fano lineshape at 90° (q→−1) around 0.9 THz (shown in Figure 2d, calculation of the Fano parameter, see method). This transition was accompanied by a π/2 phase shift from the initial position [21], demonstrating the phase-dependent nature of the resonant behavior. The simulations were conducted under plane wave excitation conditions, with an incident angle of 50°, P-polarization, and open boundary conditions within a 300 × 300 μm simulation space. The observation probe is located 2 μm from the edge of the left arm. To ensure consistency between the experimental and simulation results, the simulation parameters were carefully adjusted to match experimental conditions, which resulted in excellent agreement between the two. This validation of the observed phenomena through simulations offers critical insights into the underlying physical mechanisms governing these transitions, reinforcing our understanding of the resonator’s behavior.

Experimental near-field spectra at various rotation angles confirmed the simulations and validated the proposed mechanism for lineshape control; specifically around 0.88 THz (shown in Figure 2a,b). At 0°, we observed symmetric Lorentzian-like lineshapes; which is a typical response for systems that exhibit simple resonance. However, as the rotation angle increased, this symmetry was broken, resulting in the emergence of asymmetric Fano lineshapes. The shift in the asymmetry parameter *q* from −10 to −1 at 90° is particularly noteworthy, highlighting the spectrum-dependent nature of the transition and the pivotal role of the rotation angle in modulating the spectral response (Figure 2d red dashed line). The THz wave illuminating the tip induced a local dipolar effect, which significantly enhanced the electric field component *E*_z_ by several orders of magnitude compared to the far-field, thereby amplifying the longitudinal waveguide resonance. There is a high agreement between simulated and experimental spectral results fitted to the Fano parameter *q.*

In the time-domain signals of the U-shaped ring (shown in Figure 3), the near-field signal excited by the tip overlapped temporally with the resonant signal from the sample. As the U-shaped ring was rotated, the phase difference between these two signals varied, further emphasizing the dynamic nature of the resonance. This phase shift, which is inherently tied to the rotation angle, allowed for the precise design and quantification of the lineshapes [22]. Such capabilities are invaluable in supporting advanced applications in phase change measurements, environmental monitoring, and precision nanoscale sensors, where minute variations in resonance can be used to extract highly sensitive information from the surrounding environment.

Unlike far-field signals, which are macroscopic and lack spatial specificity, the near-field signal induced by the tip introduces a controllable element that enables asymmetric manipulation of the U-shaped ring. This ability to control the resonance at a localized level is essential for applications that require precise modulation of nonlinear characteristics. Specifically, the asymmetric excitation allows for enhanced optical switching, frequency conversion, and other nonlinear optical processes, which are critical for advancing photonic technologies. The field asymmetry induced by the tip allowed for controlled tuning of the U-shaped ring’s resonance properties [18], making it possible to achieve a wide range of resonant behaviors in a single nanoscale structure. For excitation along the boundary arm of the U-shaped ring, a π phase shift was achieved as the sample angle changed, enabling transitions from Fano to Lorentzian-like, and further to antiresonance lineshapes. This ability to manipulate resonance profiles dynamically within the same structure represents a significant advancement in resonance engineering. These transitions are visualized in Figure 4a,b, where the U-shaped ring undergoes tunable shifts based on the angle of excitation. In contrast, the opposite arm of the U-shaped ring exhibited an angle-insensitive response, particularly between 90° and 180° (shown in Figure 4c,d). This phenomenon emphasizes the spatial asymmetry within the U-shaped ring and offers further opportunities for fine-tuning the resonance behavior for specific applications.

This nanoscale resonance structure, enhanced by the underlying waveguide substrate, presents significant potential for finely tuning its resonance profile. When the substrate thickness is set to 8 µm, the resonance displays distinct Fano characteristics, as shown in Figure 5a,b. Notably, as the waveguide layer thickness increases, the intensity of the resonance also increases, causing a transition from a Fano lineshape to an antiresonance profile. This behavior indicates that the system’s lineshape is strongly influenced by the coupling between the U-shaped resonator and the waveguide layer. The coupling between the two components significantly alters the spectral characteristics, which highlights the critical role of substrate engineering in controlling the overall resonance behavior. Furthermore, the resonance is highly sensitive to variations in the refractive index of the waveguide, allowing for precise fine-tuning of the spectral response. This tunability can be achieved by adjusting the material composition. Such sensitivity to refractive index changes provides a powerful tool for tailoring the resonance properties to meet specific application requirements. Typically, the U-shaped resonator on a silicon substrate alone cannot produce a significant resonance at 0.9 THz, but when coupled with the waveguide, it creates a much more pronounced and tunable spectral response.

In addition, simulations were conducted to analyze the spectral behavior for different waveguide lengths (denoted as *L*, Figure 5c,d). The results show that as *L* increases, the resonance frequency shifts to lower values, confirming that the resonance is indeed dependent on the length of the waveguide and its associated coupling effects. This further reinforces the notion that the resonance profile is not a standalone feature of the U-shaped ring but is intricately linked to the entire integrated system. The enhanced coupling effect observed here underscores the importance of substrate engineering and waveguide design in optimizing the resonance characteristics. This combination offers a pathway to developing highly tunable photonic devices with adjustable resonance properties, enabling a range of applications in fields such as integrated photonics, sensing, and dynamic signal processing.

## 4. Conclusions

We have demonstrated the tunability between Lorentzian-like, Fano, and antiresonance lineshapes by employing phase shifts in time-domain transitions within a single U-shaped resonant ring using THz s-SNOM. Our results reveal the underlying mechanisms governing nanoscale resonant behaviors, particularly highlighting the significant asymmetry induced by near-field interactions that cannot be observed in far-field systems. The strong agreement between experimental and simulation data validates the observed lineshape transitions and underscores their critical role in modulating optical properties at the nanoscale. Similar to the effect of rotation angle, the polarization direction of the incident THz wave can linearly control the resonance characteristics. This polarization-dependent control provides an additional degree of freedom for tuning nanoscale resonances, offering new possibilities for dynamic manipulation of the optical properties.

These findings pave the way for numerous advanced applications in nanoscale photonics. By leveraging phase control in time-domain at the nanoscale, we can develop adaptive photonic devices, such as high-precision sensors capable of detecting minute environmental changes, on-chip optical signal processors, and reconfigurable photonic circuits. Moreover, the ability to precisely tune lineshapes at the nanoscale opens up exciting opportunities in fields like quantum computing, where control over qubit interactions is paramount, and biochemical sensing, where detecting subtle molecular interactions is critical.

Future work could further explore integrating this profile controlled nanoresonator into more complex photonic systems. Specifically, developing hybrid platforms that combine multiple nanoresonators with different resonant behaviors could achieve even more sophisticated spectral control. Such systems could be used for dynamic spectral tuning, offering practical solutions for advanced imaging techniques and quantum information processing. Additionally, scaling the integration of profile-controlled resonators could lead to the realization of compact, on-chip devices capable of adaptive filtering, high-speed modulation, and real-time sensing, driving innovation in next-generation photonic technologies.

## Figures and Tables

**Figure 1 nanomaterials-14-01932-f001:**
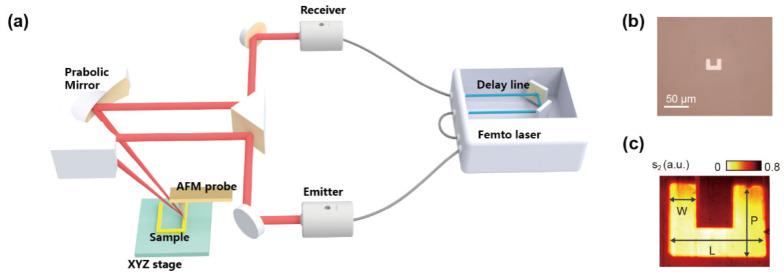
(**a**) Schematic diagram of the THz s-SNOM system. (**b**) Optical image and (**c**) second-order near-field white-light image of the U-shaped ring.

**Figure 2 nanomaterials-14-01932-f002:**
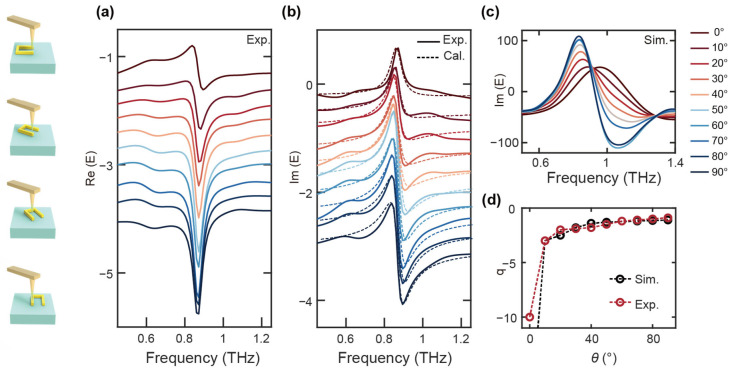
(**a**) Experimental real of the near-field spectra from 0° to 90°. (**b**) Experimental and Fano formula fitted imaginary near-field spectra from 0° to 90°. (**c**) Simulated imaginary part of the spectrum as the U-shaped ring rotates from 0° to 90°. (**d**) Fano parameters *q* of experimental and simulated imaginary parts of the spectrum.

**Figure 3 nanomaterials-14-01932-f003:**
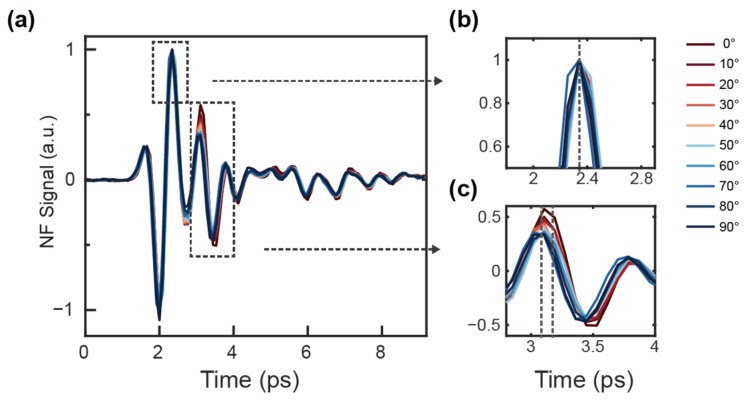
(**a**) Near-field time-domain signal from 0° to 90°, (**b**,**c**) The enlarged view of the highlighted region of (**a**).

**Figure 4 nanomaterials-14-01932-f004:**
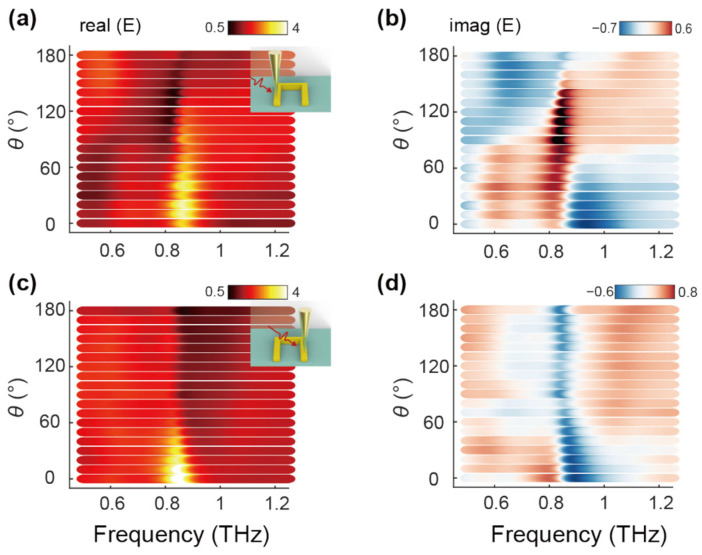
(**a**,**b**) Near-field spectra with the tip positioned in the left arm of the U-shaped ring, showing the real (**a**) and imaginary (**b**) parts from 0° to 180°. (**c**,**d**) Near-field spectra with the tip positioned in the right arm of the U-shaped ring, showing the real (**c**) and imaginary (**d**) parts from 0° to 180°.

**Figure 5 nanomaterials-14-01932-f005:**
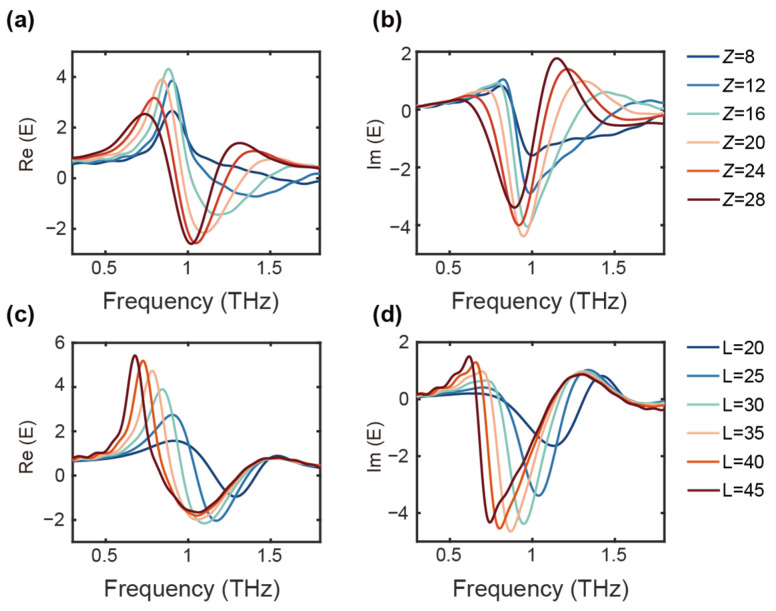
Simulated real (**a**) and imaginary parts (**b**) of the spectrum for waveguide substrate thicknesses *Z* from 8 to 28 µm. Simulated real (**c**) and imaginary parts (**d**) of the spectrum for *L* from 20 to 45 µm.

## Data Availability

Data presented in this study are available at the links mentioned in the text or on request from the corresponding author.

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
