# Peer review of "Angle-Controlled Nanospectrum Switching from Lorentzian to Fano Lineshapes"

_nanomaterials, 2024, doi:10.3390/nano14231932_

Round 1
Reviewer 1 Report
Comments and Suggestions for Authors
The authors report on the observation of the continuous change from Fano to Lorentz lineshape in the THz frequency range with nanometre-scale resolution. They employed near-field s-SNOM technique on a U-shaped micro-plasmonic resonator coupled to an underneath waveguide and achieved the spectral change y rotating the sample with respect to the optical setup.
I found the paper interesting, well-written, and containing a sufficient amount of novelty to be considered for publication. However, I identified some issues that the authors could address as minor revisions:
1) In figure 1 a) the setup schematics needs to be explained. They could define at least the important elements in the image and add a brief description in the caption. For instance, it is not clear which element is the photoconductive antenna.
2) In the acronym THz-TDS-SNOM, TDS (time-domain spectroscopy) was not defined the first time it appears in the text. It will help the reader to define it more clearly.
3) Why the simulations never show a clear Lorentzian spectral response?
4) In figure 1 c) the author should specify how large is the area from which they calculated the spectral response. The simulation procedure is briefly described later in the Results section. For a direct comparison with the experiments, I suggest moving the figure 1c) to the Results section.
5) It is not clear how large are both simulation area and experimental region. The experimental spectra of figure 2 are the average of the scan over the whole U-resonator (side length of tens of micrometers) or are detected in a single position? This point must be clearly explained. In figure 3) the spectra are obtained in a single position, when the tip is on a specific position over the resonator but in figure 1c and figure 2 this is not clear.
6) The normalization procedure of the signal lacks explanations, neither the equations that describe the process.
7) Is the spectral lineshape also depending on the resonator geometry, height for instance? Can the authors address this question at least by performing calculations?
8) I guess that when the sample rotates, there is a polarization dependance in the change of the lineshape from Fano to Lorentzian due to the different the coupling between resonant and non-resonant signals. Can the author comment about this?
9) Can the authors quantify how much “the experimental near-field spectra at various rotation angles matched well with the simulations”?
Author Response
Comments 1: In figure 1 a) the setup schematics needs to be explained. They could define at least the important elements in the image and add a brief description in the caption. For instance, it is not clear which element is the photoconductive antenna.
Response 1: Thank you very much for pointing out this issue. We have made the necessary modifications in Figure 1(a) and added names to the components of the system. For further details, please refer to Figure 1 in the Introduction section of the manuscript.
Comments 2: In the acronym THz-TDS-SNOM, TDS (time-domain spectroscopy) was not defined the first time it appears in the text. It will help the reader to define it more clearly.
Response 2: Thank you for point out this mistake in the article. We have provided the initial explanation within the manuscript, specifically in the Experiment Setup section, where the details can be found as follows:” In our experiment, we employed a commercial scattering-type terahertz time domain spectroscopy near-field scanning system (THz-TDS s-SNOM) developed by Nearspec, Germany.”
Comments 3: Why the simulations never show a clear Lorentzian spectral response?
Response 3: Thank you for pointing this out. We have increased the number of grid points in our simulations to enhance the accuracy of the results. This improvement has led to a clearer depiction of the Lorentzian line shape, thereby optimizing the reliability and precision of our findings. For further details, please refer to Figure 2(c) in the Results section.
Comments 4: In figure 1 c) the author should specify how large is the area from which they calculated the spectral response. The simulation procedure is briefly described later in the Results section. For a direct comparison with the experiments, I suggest moving the figure 1c) to the Results section.
Response 4: Thank you for your valuable suggestion. We added the corresponding description and moved figure 1c) to the Results section. For further details, please refer to Figure 2(c) in the Results section.
Comments 5: It is not clear how large are both simulation area and experimental region. The experimental spectra of figure 2 are the average of the scan over the whole U-resonator (side length of tens of micrometers) or are detected in a single position? This point must be clearly explained. In figure 3) the spectra are obtained in a single position, when the tip is on a specific position over the resonator but in figure 1c and figure 2 this is not clear.
Response 5: Thank you for pointing out the issue. In Figure 2, the experimental spectrum was detected at a single position, whereas in Figure 3, the terahertz wave was incident from the left arm, and the spectral information was obtained by scanning within a 2 µm region near the tip. For further details, please refer to the content in the Results section, which reads as follows: " The simulations were conducted under plane wave excitation conditions, with an incident angle of 50°, P-polarization, and open boundary conditions within a 300 × 300 μm simulation space. The observation probe is located 2 μm from the edge of the left arm."
Comments 6: The normalization procedure of the signal lacks explanations, neither the equations that describe the process.
Response 6: Thank you for your valuable suggestion. This is a standard practice for signal normalization in terahertz time-domain spectroscopy, as detailed in references [19] and [20].
Comments 7: Is the spectral lineshape also depending on the resonator geometry, height for instance? Can the authors address this question at least by performing calculations?
Response 7: Thank you for pointing this out. We appreciate the valuable suggestion. In response, we have added a discussion on the impact of variations in the structural parameter L on the results. For further details, please refer to Figure 5 and the corresponding text in the Results section, where the content is as follows: " In addition, simulations were conducted to analyze the spectral behavior for different waveguide lengths (denoted as ?,Figure 5c and d). The results show that as ? increases, the resonance frequency shifts to lower values, confirming that the resonance is indeed dependent on the length of the waveguide and its associated coupling effects. This further reinforces the notion that the resonance profile is not a standalone feature of the U-shaped ring but is intricately linked to the entire integrated system."
Comments 8: I guess that when the sample rotates, there is a polarization dependance in the change of the lineshape from Fano to Lorentzian due to the different the coupling between resonant and non-resonant signals. Can the author comment about this?
Response 8: Thank you for your suggestion. We acknowledge that variations in polarization have an impact on the results, and we have included a relevant discussion in the conclusion section. For further details, please refer to the content in the Conclusion section, which reads as follows: " Similar to the effect of rotation angle, the polarization direction of the incident THz wave can linearly control the resonance characteristics. This polarization-dependent control provides an additional degree of freedom for tuning nanoscale resonances, offering new possibilities for dynamic manipulation of the optical properties."
Comments 9: Can the authors quantify how much “the experimental near-field spectra at various rotation angles matched well with the simulations”?
Response 9: We are grateful for your feedback. We have added a corresponding discussion where we compare the simulation and experimental results using the fitted coupling coefficient q. The results indicate a strong consistency between the experimental and simulation results. For further details, please refer to Figure 2 d) and the corresponding text in the Materials and Methods section and Results section.
The corresponding text in the Materials and Methods section is as follows: "
Method: Calculation of Fano Resonance Line Shape
To model the Fano resonance line shape and compute its corresponding absorption cross section in terms of frequency, we start with the standard Fano formula. The Fano resonance is characterized by the interaction between a discrete state and a continuum of states, leading to an asymmetric line shape in the absorption spectrum. The absorption cross section is given by the following expression:
δFano(E)=δ0*(q+e)^2/(q+e)^2
where:E is the photon energy, q is the Fano parameter that quantifies the asymmetry of the resonance, e = (E−E0)/(ℏΓ/2) is the normalized energy, with E0 being the resonance energy and Γ the width of the resonance, σ0 is the cross section far from the resonance.”
The corresponding text in the Results section section is as follows: "There is a high agreement between simulated and experimental spectral results fitted to the Fano parameter q."

Reviewer 2 Report
Comments and Suggestions for Authors
The authors used THz s-SNOM to investigate the resonance spectrum lineshape of a U-shaped resonant ring on a waveguide substrate. The following issues should be addressed.
1. The authors wrote that the tip’s resonant frequency is 16 kHz and that it works in the tapping mode. What is the frequency of the tapping mode? Is it equal to the tip’s resonant frequency?
2. In Line 87, Figure 2 should be Figure 1.
3. How is the near-field time-domain white-light imaging in Fig. 2 obtained? Also with the s-SNOM tip? The authors should describe this in the manuscript.
Author Response
Comments 1: The authors wrote that the tip’s resonant frequency is 16 kHz and that it works in the tapping mode. What is the frequency of the tapping mode? Is it equal to the tip’s resonant frequency?
Response 1: We sincerely appreciate your thoughtful question. The mode of operation described is fundamental to near-field optical atomic force microscopy. In tapping mode, the working frequency is determined by the resonance frequency of the probe, and it is important to note that each probe exhibits slight variations in its resonance frequency. In the experiments presented in this manuscript, we performed the analysis using the scanning results obtained from a single probe to ensure consistency across the measurements.
Comments 2: In Line 87, Figure 2 should be Figure 1.
Response 2: Thank you for your insightful comment. As detailed in the "Material Preparation" section of the manuscript, the specific information you have requested can be found under the following description: “Figure 1.c illustrates a schematic of the sample as observed through an optical microscope.”
Comments 3: How is the near-field time-domain white-light imaging in Fig. 2 obtained? Also with the s-SNOM tip? The authors should describe this in the manuscript.
Response 3: Thank you for pointing this out. Near-field time-domain white light imaging is achieved by scanning the s-SNOM tip over a designated region, utilizing the time-domain peak detection imaging mode. We have explained this in the Results section of this article. the specific information you have requested can be found under the following description: “The near-field white-light imaging was conducted using peak detection imaging mode, which focuses on maximizing the signal-to-noise ratio by capturing the strongest response at each frequency.”

Round 2
Reviewer 2 Report
Comments and Suggestions for Authors
The authors adequately addressed the raised issues.